American Society for Microbiology | Microbiology Spectrum

# Tanzanian goat gut microbiomes adapt to roadside pollutants and environmental stressors

Emilie Egholm Bruun Jensen,[1] Marie Louise Jespersen,[1] Christina Aaby Svendsen,[1] Tolbert Sonda,[2] Saria Otani,[1] Frank M. Aarestrup[1]

**ABSTRACT** The impact of environmental pollution reaching and affecting the gut microbiome is rising. Pollution from vehicle emissions can release compounds harmful to both animal and environmental health, and their effect on the host microbiome is yet to be determined, particularly in understudied locations. Here, we have investigated the potential effect of environmental pollution on the gut microbiome of Tanzanian goats grazing near a heavily trafficked road compared to goats living in a more rural setting. We identified 1,468 metagenome-assembled genomes (MAGs), of which 768 were unidentified species, and created a genomic database to which 52% of the bacterial community could be assigned. We find significant differences in the composition of the bacterial communities and resistomes between rural and road-exposed goats, but not a major difference in antimicrobial resistance (AMR) abundance. Genes involved in pollutant biodegradation were significantly more abundant in the microbiome of goats grazing along the road. This includes genes involved in degradation of naphthalene and toluene (both present in motor vehicle exhaust), as well as the detoxification enzyme, glutathione S-transferase. These findings suggest living near a heavily trafficked road selects for xenobiotic degrading functions within the goat gut microbiome, which might aid the host in detoxification of these compounds.

**IMPORTANCE** To the best of our knowledge, this is the first study on the potential effect of environmental pollution on the gut microbiome of Tanzanian goats. Using shotgun metagenomics, we compare the gut microbiome of goats living near a heavily-trafficked road in Kigoma, Tanzania, with the gut microbiome of goats living in a rural area. We find that genes involved in pollutant biodegradation were significantly more abundant in the gut microbiome of the road-exposed goats, which potentially aids pollutant detoxification in the host. The effect of environmental pollution on the gut microbiome remains poorly understood; however, with this study, we link a potential effect of environmental pollution to changes in the gut microbiome of Tanzanian goats.

**KEYWORDS** metagenomics, environmental pollution, gut microbiome, genomics

**Peer Reviewer** Kirsten Zwally, Texas A&M University, College Station, Texas, USA

Address correspondence to Emilie Egholm Bruun Jensen, emegbr@food.dtu.dk.

The authors declare no conflict of interest.

See the funding table on p. 15.

The impact of environmental pollutants reaching and affecting the gut microbiome is becoming more apparent (1–7). Especially, air pollution from vehicle emissions can introduce gases and compounds harmful to both animal and environmental health. This includes carbon monoxide, nitrogen oxides, particulate matter, and volatile organic compounds, which have been shown to alter the gut microbiome and function (5, 8–10). Dasgupta et al. (11) have shown high pollution from vehicles in Dar es Salaam, Tanzania, with the worst exposure along the wind paths of highly trafficked roads, and the authors urge the adoption of continuous air-quality monitoring and effective pollution control.

In addition to affecting the environment, plastic pollution in the form of microplastics might also affect the gut microbiome (12, 13). Plastic waste has indeed been recovered

from the rumen of slaughtered urban-grazing goats in Bukavu, Democratic Republic of the Congo (DRC) (14), Amhara Region, Ethiopia (15), and Mogadishu, Somalia (16). Plastic has also been reported to be the most common foreign body in goat rumen in both Gombe State, Nigeria (17) and Addis Ababa municipality, Ethiopia (18).

The gut microbiome consists of a complex and diverse community and plays a crucial role in host health (19–21). It is involved in both nutrient metabolism, xenobiotic and drug metabolism, immune regulation, and pathogen resistance (22). Factors such as diet, genetics, environment, and geographic location shape the gut microbiome (23–25).

The goat digestive system is unique compared to humans. The stomach of goats, like other ruminants, consists of four compartments: the rumen, the reticulum, the omasum, and the abomasum (also referred to as the true stomach) (26). As ruminants, goats are highly dependent on their gut microbiome. Similar to other vertebrates, they do not produce fiber-degrading enzymes themselves, which is essential for digestion and nutrient absorption of fibrous plant material (27), thus they rely largely on their gut microbiomes for fiber breakdown. Microorganisms, primarily within the rumen, ferment fibrous plant material to produce vitamins and volatile fatty acids, which are the primary source of energy to the host (28, 29). Generally, rumen microorganisms can be characterized based on their functionality, e.g., cellulolytic (cellulose degradation), hemicellulolytic (hemicellulose degradation), pectinolytic (pectin degradation), amylolytic (starch degradation using ammonia and amylase), proteolytic (protein hydrolysis), and lipolytic (lipid hydrolysis) (29, 30).

Previously, studies have examined the intestinal microbiome composition in goats on different diets (30), and the gut microbiome composition of goats of different ages (31), both using 16S rRNA sequencing. Recently, a metagenomic shotgun-sequencing data set was published. Here, Ma et al. (32) obtained 797 metagenomic-assembled genomes (MAGs) recovered from the rumen of 43 Laiwu black during early life. Additionally, Zhang et al. (25) have presented a gene catalog of >82 million non-redundant predicted genes from 110 fecal samples collected from Chinese farm goats. They compared them to 210 fecal samples from Chinese sheep and found a significant difference in glycan degradation and utilization patterns between sheep and goats. The rearing environment (grazing or controlled drylot) was shown to affect the composition of the gut microbiome composition and functionality in goats. In addition, goats in a controlled drylot environment had an increased abundance of pathogenic bacteria. They found that the most abundant bacterial species in the goat gut microbiome were *Escherichia coli*, *Campylobacter* sp. RM8964, *Butyricicoccus pullicaecorum*, *Campylobacter* sp. RM12175, and *Clostridioides difficile*. However, no data are available on the microbiome of goats from Africa or the potential impact of environmental factors on the composition.

Interaction between environmental pollutants and the gut microbiome is generally understudied (6). However, Lindell et al. (7) define six types of interactions between xenobiotic (ingested) compounds and the gut microbiome: growth inhibition, growth promotion, change in natural bacterial metabolites, impact on virome and virulence, xenobiotic metabolism, and xenobiotic bioaccumulation. Xenobiotics can, using these modes, directly select and shape the composition of the gut microbiome. In humans, it has been shown that gut bacteria can metabolize xenobiotics like therapeutic drugs (33, 34), industrial chemicals (35), and environmental pollutants (7, 35). The human gut bacteria use distinct chemical strategies like hydrolysis and reduction reactions, unlike those of the host metabolism which rely mainly on oxidation and conjugation, to metabolize xenobiotics (35). This microbial metabolism of pollutants can support the host in the detoxification of these compounds (36).

Little is known about the effect of environmental pollution on the gut microbiome of ruminants, e.g., goats. However, rumen bacteria are known to degrade and detoxify natural toxins found in animal feed (37). For example, the mycotoxin Ochratoxin A (OTA) produced by *Aspergillus* and *Penicillium* is a natural contaminant in plant feed in both tropical, subtropical, and temperate regions (37, 38). Carboxypeptidase A enzymes from *Bacillus licheniformis* have been linked to OTA degradation in native Korean goats (39).

Here, we utilize shotgun metagenomic sequencing to further understand the functionality of the goat gut microbiome. We aimed to investigate the effects of environmental pollution, such as vehicle emissions and plastic pollution from the main road north of Kigoma, Tanzania, on the bacterial composition and function of the gut microbiome of goats living near this busy road, compared to goats from a more rural and less polluted area.

## MATERIALS AND METHODS

### Samples

All samples were collected on the 27th of January 2024, at different locations along an approximately 20 km distance along the main road north of Kigoma, Tanzania. Either as close to the road as possible (maximum 10 m away), referred to as the road samples here, or in smaller settlements a minimum of 500 m away from the main road where it was necessary to walk the last minimum 100 m, here referred to as the rural samples. To reduce the risk of collecting multiple times from the same goats, samples were collected a minimum of 10 m apart along the road, where the goats were individually tied. In the rural area, only one sample was collected per settlement/group. Feces were scooped from the center or top of fresh fecal piles (visibly wet) to avoid soil contamination, as well as to ensure that the microbiome members did not change due to incubation on the road and collected in sterile and clean plastic tubes. A total of 15 samples were collected from a rural setting, and 19 near the road. The samples were kept as cool as possible on cooling elements until a freezer was available, and then kept frozen at 20°C until DNA extraction.

### DNA extraction, library preparation, and sequencing

Prior to DNA extraction, to eliminate potential external contamination, the surface from each fecal sample was discarded and only the untouched core part of each fecal sample was collected and used for DNA extraction. Total DNA was extracted from all core fecal samples using the QIAamp Fast DNA stool mini kit (Qiagen, Germany) following the manufacturer's instructions using 200 mg as starting material. The DNA was eluted in 50 µL of pre-heated (65°C) AE buffer to increase DNA yield. DNA quality was checked using a Qubit Fluorometer (Thermo Fisher Scientific). For metagenomics sequencing, all libraries were prepared using the PCR-free Kapa Hyper Prep Kit (Roche). All libraries were sequenced on Illumina Novaseq 6000 S4 (2 × 150 bp) platform.

### Quality control of sequencing reads

FastQC version 0.11.5 (40) was used to check the quality of the raw reads. Trimming was performed with BBduk2 version 36.49 (41) to remove adaptors, and right-end trim was applied to remove bases with Phred scores below Q20. Lastly, reads shorter than 50 bp were removed.

### Assembly and binning

*De novo* assembly of the trimmed reads was performed with SPAdes version 3.14.0 (42) using the metagenomic flag (--meta). The average depth was obtained with jgi_summarize_bam_contig_depths from MetaBat version 2.12.1 (43) with the flag -outputDepth. Contigs with length >1,500 were used for binning with MetaBat. Given the limited research and scarcity of samples from wild goat gut microbiomes, we included both high-quality (HQ) bins (completeness > 90, contamination < 5) and medium-quality (MQ) bins (completeness ≥ 50, contamination < 10). Including both HQ and MQ bins would retain important microbial data. The bins were identified with CheckM lineage_Wf version 1.0.18 (44) and dereplicated with dRep version 2.6.2 (45) with a primary ANI cluster threshold of 0.9 (-pa 0.9), and a secondary ANI cluster threshold of 0.97 (-sa 0.97).

## Taxonomic annotation of the MAGs

Taxonomic annotation of the dereplicated MAGs was performed with GTDB-Tk classify_wf version 2.3.2 with database version R207 (46). These were placed within the reference tree, and the backbone bac120 tree was visualized in R version 4.3.0 with ggtree version 3.8.2 (47).

## Read-based mapping to databases

The trimmed reads were mapped and aligned with KMA version 1.4.2 (48) with the following flags: -mem_mode -ef -1t1 -cge, against the resistance gene database PanRes version 1.0.0 (49). PanRes consists of the following databases: ResFinder (downloaded 2023-01-20 [50]), ResFinderFG version 2.0 (51), the "CsabaPal" collection (also known as ResFinderNG, provided by Csaba Pál and Zoltán Farkas in November 2022 [52]) and BacMet version 1.1 (53). The abundance table containing the aligned fragment counts was normalized by the reference genome length divided by 1,000, i.e., count/(ref_length/ 1,000). This accounts for longer references having more fragments assigned than shorter ones. A custom genomic database consisting of the dereplicated MAGs (each MAG was padded with 800 characters of lowercase $n$ to optimize KMA conclave scoring for resolving multiple template matches), bacteria (closed genomes), and bacteria_draft (draft genomes) downloaded from NCBI GenBank (54) (both downloaded 26.04.2022) was created. The trimmed reads were mapped and aligned with KMA. A mitochondrial database was downloaded from NCBI RefSeq (55) (downloaded 02.07.2022) to confirm the host. The trimmed reads were mapped and aligned against it with KMA.

## Alpha diversity measures

Bacterial alpha diversity, evenness, and richness were calculated based on the abundances from the read-alignment to the custom genomic database consisting of MAGs, bacterial, and bacterial draft reference database (as described earlier). The skbio.math.diversity.alpha python package (56) was used to calculate chao1, Simpson diversity index, and Shannon' evenness.

## Functional annotation

Prodigal version 2.6.3 (57) was used to perform gene prediction for the contigs, using the metagenomic flag (-p meta). Functional annotation of the predicted proteins was performed with the eggNOG-mapper tool version 2.1.12 (58) utilizing DIAMOND version 2.1.8 (59) for sequence alignment (evalue ≤ 0.001, score ≥ 60, pident ≥ 40, query_cover ≥ 20, subject_cover ≥ 20). If two or more KEGG Orthologs (KOs) were identified for a gene product in the output, it was split and each of the KOs counted once per gene product, since a gene product can be involved in different metabolic pathways. The trimmed reads were mapped to the gene products with BBMap version 38.90 (41), and converted from .sam to .bam with samtools version 1.20 (60). Abundance tables were created by amalgamating the per-sample output from samtools idxstats.

The ability of the gut microbiome to break down plastic was investigated with the curated Plastic Biodegradation Database (PlasticDB) (61). The 219 proteins with plastic-degrading capabilities were downloaded from https://plasticdb.org/downloaddata (data retrieved: 09.10.2024). The predicted proteins were mapped against PlasticDB with DIAMOND blastp version 2.1.8 (59) with e-value <1 e-20 and pident >30.

## Quantification of relative abundance

Relative abundance of antimicrobial resistance genes, functional genes, and plastic degrading enzymes was quantified as fragments per kilobase million (FPKM) described in equation 1. Here, $Gene_{Fragments}$ is the number of fragments assigned to the reference. $Gene_{Length}$ is the length of the reference, and $Bacteria_{Depth}$ is the number of fragments assigned to any taxonomic level from the bacterial reference database in the given sample.

$$\text{Relative abundance} = \frac{\text{Gene}_{\text{Fragments}}}{\text{Gene}_{\text{Length}} \cdot \text{Bacteria}_{\text{Depth}}} 10^9. \tag{1}$$

## Compositional data analysis

To describe the beta-diversity, bacterial genus and species abundance tables were used. The Python package pyCoDaMath (62) was used to perform zero replacement, followed by a centered log-ratio (clr, equation 2) transformation:

$$x = [x_1, \ldots, x_D], \tag{2}$$

$$\text{clr}(x_1, \ldots, x_D) = \left( \log\left( \frac{x_1}{G(x)} \right), \ldots, \log\left( \frac{x_D}{G(x)} \right) \right),$$

$$\text{where } G(x) = \sqrt[D]{x_1 \cdot, \ldots, \cdot x_D}.$$

K-means clustering was performed on the clr-transformed genera counts with the Python clustering package from Scikit-learn version 1.3.2 (sklearn.cluster.KMeans) (63) and hierarchy clustering was performed with the Python clustering package from SciPy version 1.10.1 (scipy.cluster.hierarchy) (64).

## Statistical analyses

ALDEx2 version 1.34.0 (65) was used to identify differential abundant features. Briefly, a Welch's $t$-test followed by a Benjamini-Hochberg false-discovery rate (FDR) correction was used to identify the differences in the clr abundance between the road and rural goats. Statistically significant features with absolute effect size >0.8 and FDR <0.05 were reported for each group. Effect size is calculated as the median difference in clr values between groups divided by the median of the largest difference in clr values within groups, i.e., diff.btw/max(diff.win).

The two groups (rural and roadside) were compared in terms of alpha-diversity metrics, total antimicrobial resistance (AMR) load, and the relative abundance (FPKM) of selected genes. A standard $t$-test was used when the assumption of equal variances was confirmed by Levene's test, otherwise a Welch's $t$-test was applied.

## Taxonomic annotation of contigs

Taxonomic annotation of the contigs containing relevant gene orthologs was performed with MMseqs2 taxonomy (66, 67) with the lowest common ancestry (2bLCA) approach (--lca-mode 3) against the GTDB database version R220 (68). Only annotations with high confidence were considered (e-value < 1e-50). The Python library Plotly version 5.14.0 (69) was used to create the Sankey diagrams for visualization of the taxonomy.

## Data visualization

Unless otherwise stated, all figures were created using the Python libraries Matplotlib version 3.9.2 (70) and Seaborn version 0.13.2 (71).

## RESULTS

### Sequencing, gene, and taxa alignment output

From the 34 goat gut microbiomes, 1.72 billion paired-end (PE) reads were obtained (on average, 50.60 million PE reads per sample, with a range of 41.89–107.80 million PE reads, and standard deviation of 13.39 million).

On average, 0.19% of the reads aligned to genes from the PanRes database (of which 1.66% aligned to genes from the ResFinder database, 4.06% to the ResFinderFG database, 10.64% to the ResFinderNG database (CsabaPal collection), and 10.64% to the BacMet database).

Prior to the addition of the MAGs to the genomic database, only 6.92% of the trimmed reads aligned. However, on average, 51.83% of the trimmed reads aligned per sample to the custom genomic database consisting of the dereplicated MAGs, closed bacterial genomes, and draft genomes from NCBI Genbank. A total of 51.47% of the aligned reads were annotated as belonging to any taxonomic level of the superkingdom Bacteria. The aligned per-sample fragment counts to each of the databases and the percent of reads aligned to the databases are shown in Fig. S1 and Table S1, respectively.

## Metagenomic assembly

After assembly, 798 HQ bins and 1,452 MQ were identified with checkM. Dereplication with dRep reduced these to 1,468 MAGs. The taxonomy of these MAGs at the phylum level can be seen in Fig. S2A. The MAGs were not distributed throughout the tree of known bacterial species (Fig. 1A), but grouped into specific clades. A total of 768 of the MAGs were unknown species, as they did not have an already existing reference genome (no GTDB-Tk annotation at the species level) (Fig. S2B). Of these unidentified MAGs, 374 were of high quality, and 394 were of medium quality. The majority of the unidentified species belong to the Clostridia class (Fig. 1B).

## Bacterial community

Across the 34 goat gut microbiomes, 4,146 unique bacterial species were identified with read-mapping to the custom genomic database consisting of bacteria and draft bacterial genomes from NCBI, as well as the de-replicated 1,468 MAGs from these microbiomes. The most abundant bacterial genera (calculated as clr) were *Faecousia*, *Cryptobacteroides*, *Alistipes*, *Phocaeicola*, UBA2862, and *Escherichia*. The 50 most abundant bacterial species are shown in Table S2, and the 20 most abundant bacterial genera in Table S3. All samples had *Capra* (goat) as the most abundant genus from the Metazoa kingdom using the mitochondrial database (Fig. S1G).

Unsupervised hierarchical clustering of the clr-transformed genera counts showed four samples clustering differently than the rest (Fig. S3A). K-means clustering with k = 2 also clustered these four samples together, leaving the rest of the sample in the second cluster. These four samples with different bacterial compositions were Rural_26, Rural_28, Rural_32, and Rural_37 (Fig. 2A and B). Differential abundance analysis showed 41 genera as being compositionally more abundant in these four microbiomes compared to the rest of the microbiomes, and 32 genera being more abundant in the rest of the samples compared to these four distinctive variants (Fig. S3B). Analysis on the taxonomic species level (Fig. S3C) shows that some of the bacteria driving these distinct microbiomes are opportunistic or pathogenic, e.g., *Klebsiella pneumoniae,* which can cause caprine mastitis (72, 73), some are part of healthy gut microbiomes, e.g., *Ruminococcus* sp. (74), while others are associated with soil and plant bacteria, e.g., *Bacillus* sp. (75) and *Arthrobacter* sp. (76). This suggests that the different microbiome compositions might be due to illness of the goat or soil contamination of the sample. The following analyses were performed without the four distinctive variants, since we do not believe these are representative of the typical goat gut microbiome. However, all analyses were repeated on the entire sample collection and are provided in the supplementary material.

No significant difference in the bacterial diversity (at the genus level) was detected between the rural and road goat microbiomes (Fig. S4A through C). However, lower diversity and evenness were observed in the distinctive microbiomes, as well as a higher richness (Fig. S4D through F).

Differential abundance analysis identified bacterial genera driving both the road and rural goats (Fig. 2C), with *Nanosyncoccus*, Rug563, and UBA1067 being the most differentially abundant in the road goat microbiomes, and UBA1723, *Butyrivibrio*, and *Delftia* as the three differentially abundant genera in the rural ones.

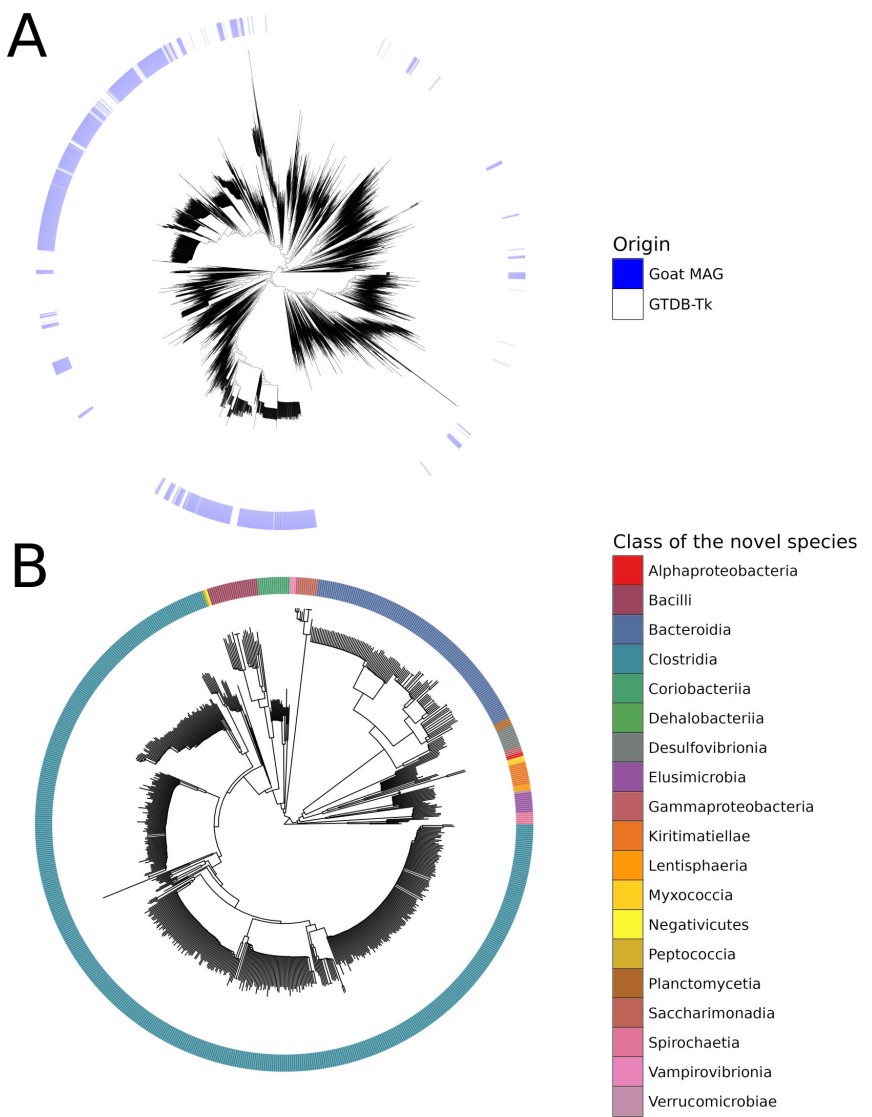

Origin
■ Goat MAG
□ GTDB-Tk

Class of the novel species
■ Alphaproteobacteria
■ Bacilli
■ Bacteroidia
■ Clostridia
■ Coriobacteriia
■ Dehalobacteriia
■ Desulfovibrionia
■ Elusimicrobia
■ Gammaproteobacteria
■ Kiritimatiellae
■ Lentisphaeria
■ Myxococcia
■ Negativicutes
■ Peptococcia
■ Planctomycetia
■ Saccharimonadia
■ Spirochaetia
■ Vampirovibrionia
■ Verrucomicrobiae

**FIG 1** Taxonomy of the goat MAGs. (A) Location of the goat MAGs within the GTDB-Tk reference tree of known bacterial species. (B) Taxonomy of the unidentified/novel goat MAG species (no existing reference genome).

## Resistome variations

The most abundant AMR genes from the ResFinder database were tetracycline genes (*tet(w)_5_aj427422*, *tet(q)_2_x58717*, and *tet(o/32/o)_6_ng_048124*) in the road goats, and tetracycline and sulfonamide genes *(tet(w)_5_aj427422*, *tet(o/32/o)_6_ng_048124*, and *sul2_2_ay034138*) in the rural goats. The top 20 most abundant AMR genes for rural and road goats from the ResFinder database are shown in Table S4.

The four previously identified distinct microbiomes with different bacterial compositions (Rural_26, Rural_28, Rural_32, and Rural_37) also had different resistome patterns with less tetracycline resistance and more resistance toward folate pathway antagonists (Fig. 3C) and a significantly higher AMR load: Rural_32 (FPKM = 307), Rural_26 (FPKM = 240), Rural_28 (FPKM = 224), and Rural_37 (FPKM = 160) (Fig. S5). When removing the four distinct microbiomes, no difference in AMR load was observed for the resistance genes in the ResFinder database (Fig. 3B); however, clustering was to some degree still observed in the ordination analysis (Fig. 3A), with *tet(q)_2_x58717* being differentially abundant and driving the road goat microbiomes.

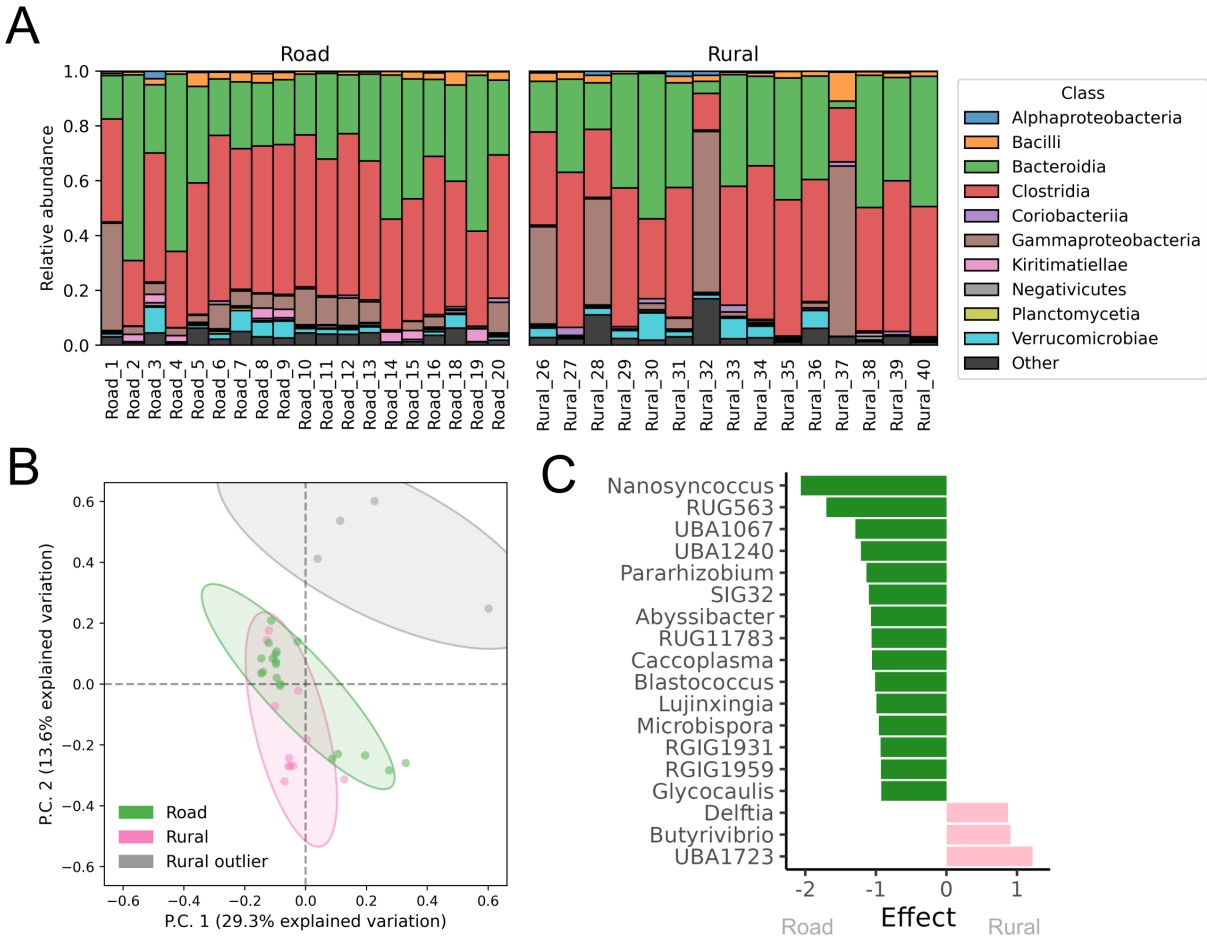

**FIG 2** Differences in the bacterial community for Tanzanian rural and road goats. (A) Relative abundance of bacterial classes calculated as FPKM. (B) Bacterial genera clustering for road and rural Tanzanian goats. Features with clr median >−1 and clr variance >2 were included. (C) Differential abundant bacterial genera. The distinctive variants (Rural_26, Rural_28, Rural_32, and Rural_37) were not included. Analysis of the distinct variants versus the rest of the samples can be seen in Fig. S3B and S3C, and analysis of all rural vs all road samples can be seen in Fig. S3D. Statistically significant features with absolute effect size >0.8 and FDR <0.05 are shown.

Further analysis of the resistome by read-mapping to the resistance genes identified with functional metagenomics (ResFinderFG) and to novel resistance genes (ResFinderNG) showed increased AMR load in the road samples (Fig. S6 and Fig. S7). The most abundant resistance genes for these databases can be seen in Table S5 and Table S6. Analysis of metal resistance genes (BacMet database) yielded similar results: The road samples had a significantly higher metal resistance load (Fig. S8). The most abundant metal resistance genes are shown in Table S7.

## Plastic biodegradation capabilities within the gut microbiome of Tanzanian goats

Across the 34 goat gut microbiomes, 181 unique proteins with plastic-degradation capabilities were identified. The most common plastic types that enzymes were capable of degrading were PBS-PBSA-PCL, 3HV-PHBV-PHA, and PEG in the road goat guts and P3HV-PHBV-PHA, PLA, and PEG in the rural goats (Table S8). The top 20 most abundant plastic-degrading enzymes for road and rural goats can be seen in Table S9. No difference in the composition of the plastic-degrading enzymes was observed (Fig. S9A through C), and no enzymes were differentially abundant between the two groups. There was also no difference in the total abundance of plastic-degrading capabilities calculated as FPKM (Fig. S9B through D).

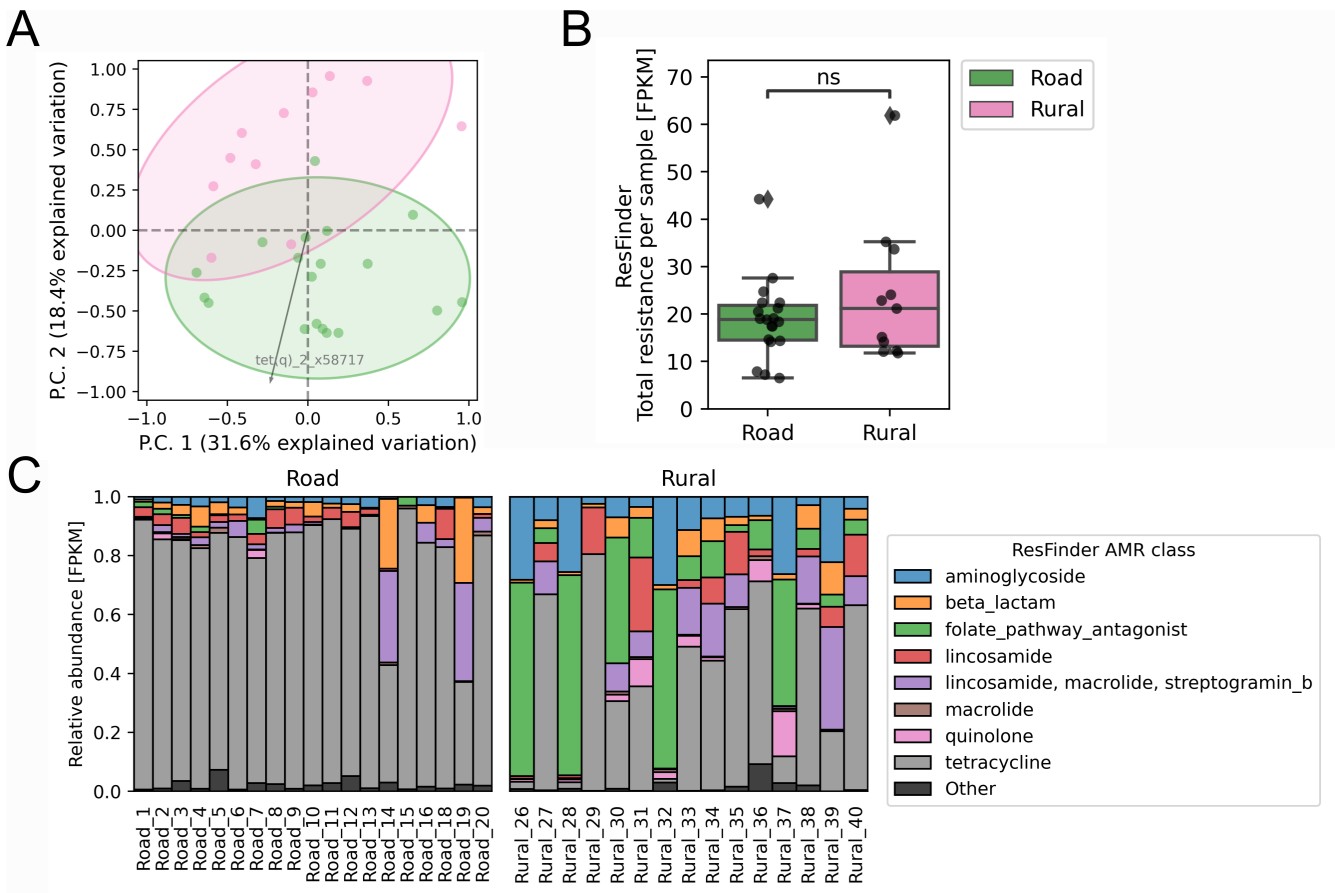

**FIG 3** Resistome differences in road and rural Tanzanian goats based on read-mapping to the ResFinder database. (A) Clustering of AMR genes from the ResFinder database. The four distinct variants with a different bacterial composition and a high total antimicrobial resistance were not included (Rural_26, Rural_28, Rural_32, and Rural_37). Only the differential abundant *tet(q)_2_x58717* is shown and is driving the road goats (effect = 1.468). (B) Total AMR load calculated as FPKM (independent *t*-test, $P = 0.233$). *$P < 0.05$, **$P < 0.01$, ***$P < 0.001$. The four distinct variants were not included (Rural_26, Rural_28, Rural_32, and Rural_37). (C) Relative abundance of the resistome for road and rural goats calculated as FPKM.

## The gut microbiome of road goats is abundant in xenobiotic-degrading functions

On average per sample, 210,579 gene products were predicted from the contigs with Prodigal. Out of these, an average of 4,954 unique gene products were annotated by eggNOG per sample (average non-unique total per sample: 19,359). Differences in the molecular functions were investigated with KEGG. Initially, no differential abundant pathway maps were identified (Fig. S10). However, when removing the four previously identified distinctive, potentially infected, microbiomes (Rural_26, Rural_28, Rural_32, and Rural_37), differential abundant pathways were found driving the two groups (Fig. 4D).

Pathway maps from the KEGG group *09111 Xenobiotics biodegradation and metabolism* were found to be enriched in the gut microbiome of the road goats. This group of pathways is capable of degrading various xenobiotics, which are chemical compounds foreign to the host, e.g., pollutants. Interestingly, naphthalene and toluene degradation were significantly enhanced in the road goat gut functions (Fig. 4A and B). Both naphthalene and toluene are found in fugitive emission and motor vehicle exhaust (77, 78). Nine unique functional orthologs were recovered from the naphthalene degradation pathway map (Fig. S11A), with three functional orthologs, K14582 (cis-1,2-dihydro-1,2-dihydroxynaphthalene/dibenzothiophene dihydrodiol dehydrogenase), K04072 (acetaldehyde dehydrogenase/alcohol dehydrogenase), and

K00480 (salicylate hydroxylase) being significantly enriched in road goats (Fig. S12A). Seven unique functional orthologs were recovered from the toluene degradation pathway map (Fig. S11B), with two functional orthologs being significantly enriched in road goats: K03268 (benzene/toluene/chlorobenzene dioxygenase subunit alpha) and K18089 (benzene/toluene/chlorobenzene dioxygenase ferredoxin component) (Fig. S12B).

Pathways related to cancer were also differentially abundant in road goats (Fig. 4C). For all of these (chemical carcinogenesis - receptor activation [PATH:ko05207], chemical carcinogenesis - DNA adducts [PATH:ko05204], pathways in cancer [PATH:ko05200], and hepatocellular carcinoma [PATH:ko05225]), the differential abundance observed was due to the high abundance of functional ortholog K00799 (Fig. S12G). This is glutathione S-transferase (GST), which is a group of phase II detoxification enzymes (79, 80). High abundance of GST was also the reason for other differentially abundant pathways (drug metabolism - other enzymes [PATH:ko00983], fluid shear stress and atherosclerosis [PATH:ko05418], and platinum drug resistance [PATH:ko01524]) (Fig. S12G). GST was also present in the glutathione metabolism [PATH:ko00480] together with three other recovered unique functional orthologs (K04097, K00310, K21888). However, of the four functional orthologs, only the GST was significantly enriched in road goats (Fig. S12E). This was also true for drug metabolism - cytochrome P450 [PATH:ko00982], where two

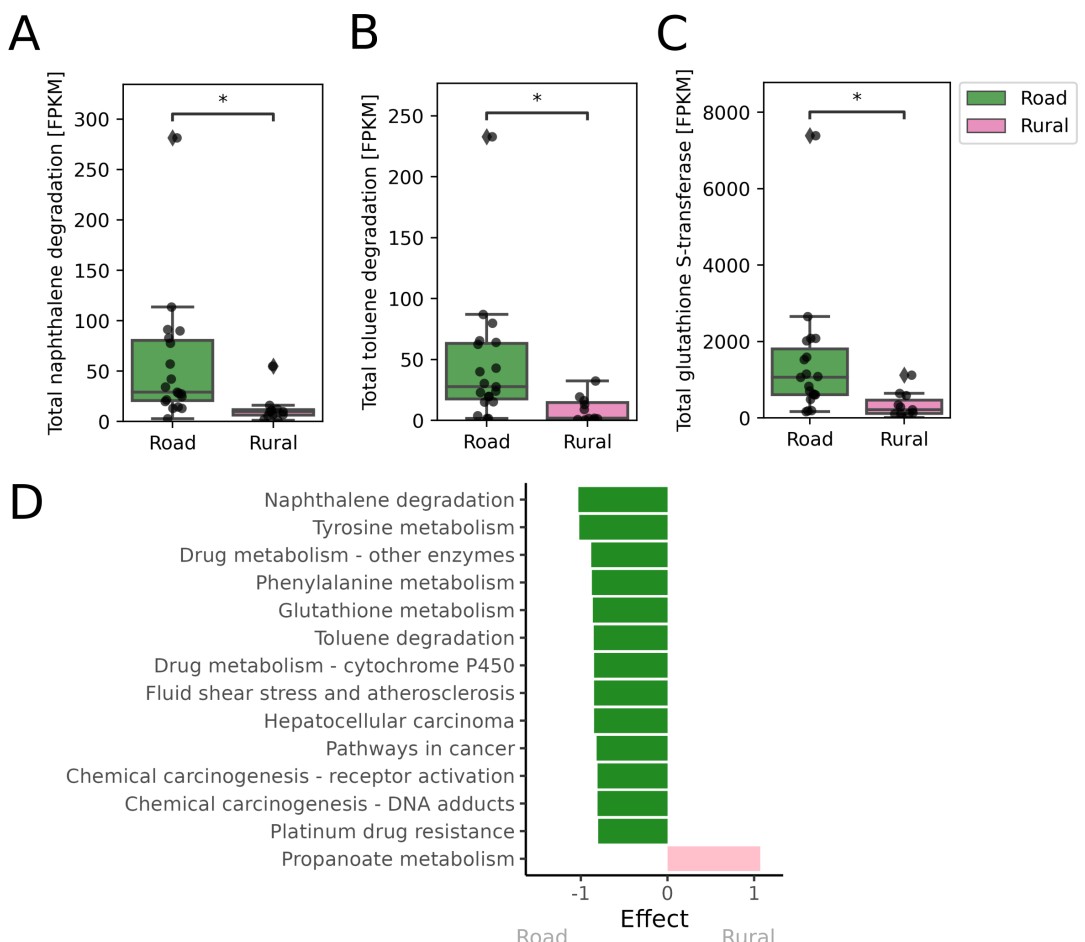

**FIG 4** Differences in the functional microbiome. The four distinct variants were not included (Rural_32, Rural_26, Rural_28, and Rural_37). (A) Total naphthalene degradation [PATH:ko00626] calculated as FPKM (independent $t$-test, $P$ = 3.375e-02). (B) Total toluene degradation [PATH:ko00623] calculated as FPKM (independent $t$-test, $P$ = 3.213e-02). (C) Total glutathione S-transferase (K00799) calculated as FPKM (independent $t$-test, $P$ = 3.688e-02). (D) Differential abundance analysis of all KEGG pathway maps. Statistically significant features with absolute effect size >0.8 and FDR <0.05 are shown. *$P$ < 0.05, **$P$ < 0.01, ***$P$ < 0.001.

unique functional orthologs were recovered (K00799, K04097), with only GST being significantly enriched in road goats (Fig. S12G).

Tyrosine metabolism [PATH:ko00350] was found to be differentially abundant in road goats. Nine unique functional orthologs (K00146, K00151, K02509, K16165, K00450, K04072, K00817, K01800, K01801) were recovered from tyrosine metabolism (Fig. S11C), with eight being significantly enriched in the road goats (Fig. S12C). Phenylalanine metabolism [PATH:ko00360] was also differentially abundant in road goats. Here, 13 unique functional orthologs (K02615, K00146, K05711, K05714, K05708, K00817, K02554, K05710, K01692, K01666, K00529, K18359, K05712) were recovered (Fig. S11D), with seven being significantly enriched in the road goats (Fig. S12F). Propanoate metabolism [PATH:ko00640] was the only pathway being differentially abundant in rural goats. Three functional orthologs were recovered (K13922, K01692, K20455), with only propionalde-hyde dehydrogenase (K13922) being significantly enriched in rural goats (Fig. S12D).

## Linking gene orthologs to taxa

Taxonomic assignment of contigs containing naphthalene-degrading gene orthologs, as well as toluene-degrading gene orthologs and the gene ortholog K00799, i.e., glutathione S-transferase, was attempted. The identified gene orthologs involved in naphthalene degradation were found on 257 gene contigs, of which 150 were from road goat microbiomes. Eighty-two contigs contained gene orthologs involved in toluene degradation, of which 53 were from road samples. None of these contigs were part of a HQ or MQ bin. Gene ortholog K00799 was located on 612 contigs, of which 458 were from road samples, of these one contig was part of a bin. This bin, originally from sample Road_8, was annotated *E. coli* by GTDB-Tk.

Instead, taxonomic annotation of the contigs from the road samples was performed with a lowest common ancestor (LCA) approach (see Materials and Methods). The majority of the gene orthologs involved in naphthalene degradation were originating from *Escherichia* genus, and for gene orthologs in toluene degradation, all of the road contigs were *Escherichia* genus (Fig. 5A and B). Both *Congzhengia* sp. (5 contigs), DTU089 sp. (3 contigs), and *Coprovivens* sp. (3 contigs) were also common. The majority of contigs containing the Glutathione S-transferase (K00799 gene ortholog) were from *Escherichia* genus (73 contigs), and *Klebsiella* (9 contigs), *Oxalobacter* (9 contigs), JAFUYI01 (5 contigs), RUG410 (11 contigs), and CAZU01 (6 contigs) were common genera (Fig. 5C).

## DISCUSSION

To the best of our knowledge, this is the first study investigating the gut microbiome of Tanzanian goats and the potential effect of environmental pollution on their microbiome. Utilizing shotgun metagenomics, we have characterized the composition of the bacterial community of Tanzanian goats. Prior to assembly and binning, we were only able to assign 7% of the trimmed reads to taxa, which emphasizes the large amount of unknown bacterial taxa and the need to characterize these. Microbiome studies in sub-Saharan Africa are, in general, under-represented compared to the rest of the world (81, 82), and this is especially true for the African livestock, where microbiome studies are still lacking (83).

After assembly, binning, and dereplication of medium and high-quality bins, we obtained 1,468 MAGs, with 768 of them being unidentified species mainly from Clostridia class. Of these 768 MAGs with no species annotation, 374 were of HQ, suggesting that we have recovered potential novel microbial species. Assembling our goat microbiomes, while time-consuming and computationally demanding, increased bacterial identification and assignments to 52% when including the new MAGs in our genomic database. A recent shotgun metagenomic-based study (25) comparing the gut microbiome of sheep and goats in China found that the most abundant bacterial species in the goat gut microbiome was *E. coli*, different *Campylobacter* species, as well as *Butyricicoccus pullicaecorum* and *Clostridioides difficile*. While the most abundant bacterial genera in the Tanzanian goat gut microbiome from this study were *Faecousia*, *Cryptobacteroides*,

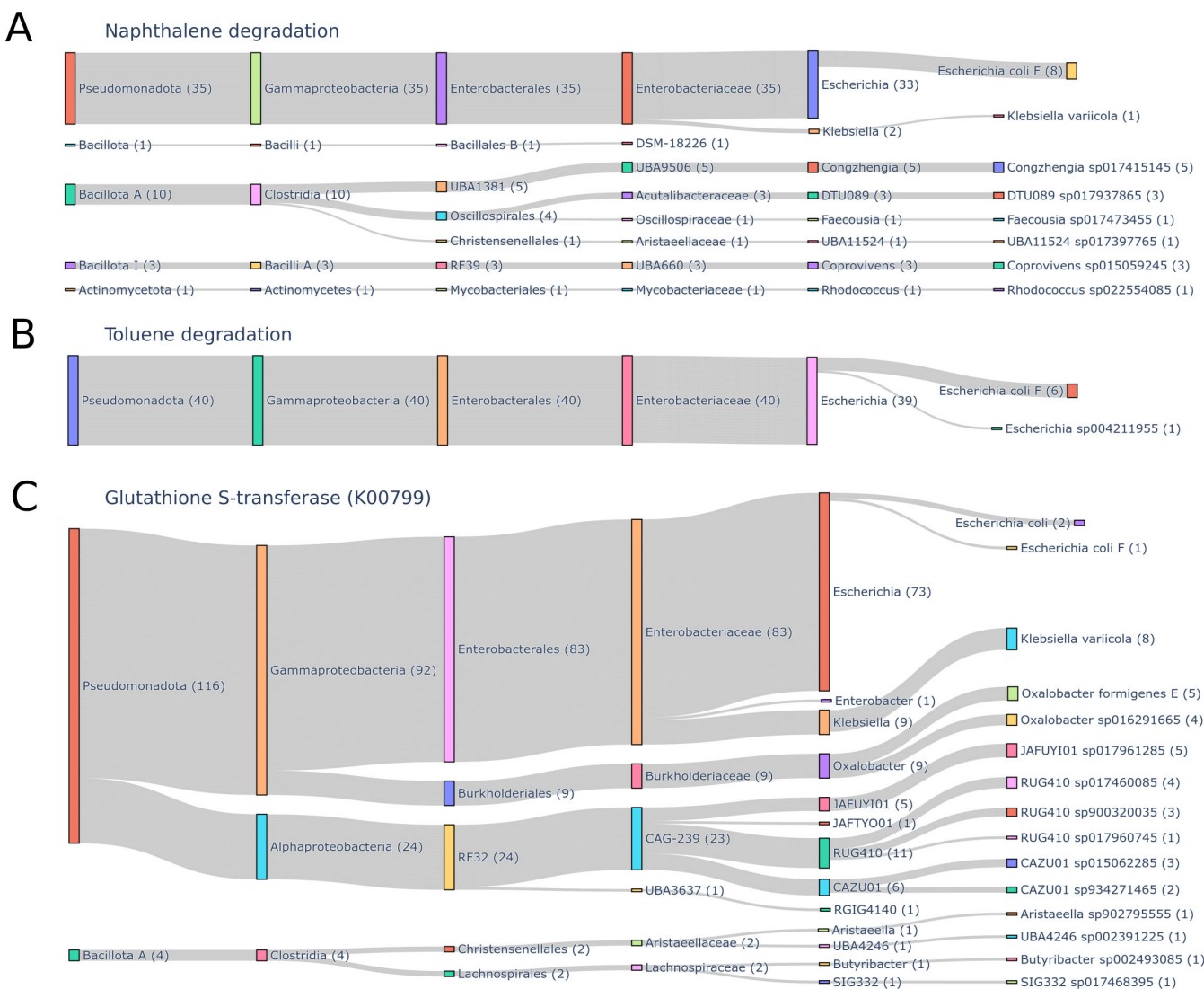

**FIG 5** Linking road contigs containing specific gene orthologs to taxa. The number of contigs is written in parentheses after the taxa. (A) Taxonomy of the contigs containing gene orthologs involved in naphthalene degradation. (B) Taxonomy of the contigs containing gene orthologs involved in toluene degradation. (C) Taxonomy of the contigs containing gene ortholog K00799, i.e., glutathione S-transferase.

*Alistipes*, *Phocaeicola*, UBA2862, and *Escherichia*. These variations might be explained by the Chinese goats being bred in farms, as well as their diet and geographic location, all of which are factors known to shape the composition of the gut microbiome (23, 84, 85).

Four rural goat gut microbiome samples had different bacterial compositions compared to the rest of the samples. The host was confirmed to be goat by read-mapping to a mitochondrial database, eliminating the possibility of a different vertebrate host. Bacteria from the Gammaproteobacteria class were observed to be compositionally increased in the five gut microbiomes (the four mentioned above plus sample Road 1). However, at the genus level, this difference was only observed in the four rural samples: unsupervised clustering at this taxonomic level showed that these four samples clustered separately from the rest of the samples. A differential abundance analysis was conducted to identify which bacterial taxa were driving the difference between the four distinct microbiomes and the rest of the samples. Differentially abundant bacterial taxa in these four microbiomes suggested that these four goats might suffer from a GI infection, e.g., *Klebsiella pneumoniae*, known to cause caprine mastitis (72, 73). Additionally, *Corynebacterium* genera was also found to be differential abundant in these four

microbiomes. *Corynebacterium pseudotuberculosis* causes caseous lymphadenitis in goats (86, 87), and it can be speculated that they might suffer from this infectious disease. Another reason for the observed clustering could be contamination, since both soil- and plant-associated bacteria were found to be differentially abundant in the four distinct microbiomes. Although we took measures to avoid contamination during collection and DNA extraction (see Materials and Methods), we cannot entirely exclude this possibility. These four distinct rural goat microbiomes also differed in their resistance patterns and had a high AMR load. This could be explained by the presence of pathogens, e.g., from the Gammaproteobacteria class, which might have higher numbers of AMR genes that are well represented in the database. Another reason could be that these four goats might have had antibiotics treatment; however, since no information on treatment status could be obtained for any of the goats, this could not be confirmed. Lastly, Gammaproteobacteria blooms have been shown to happen in fecal samples at room temperature, which might confuse the microbiome composition result (88). This could also be an explanation of the increased Gammaproteobacteria observed in these four samples. The diet of these four goats could also have been different. Most Tanzanian ruminants depend on natural pasture as their feed (89), which is also true for, e.g., dairy goats. Besides grazing on natural pastures, the goat might also feed on nearby crop residues, trees, and scrub (90, 91). However, since no diet information is available on either of the goats, this cannot be confirmed. Regardless of the reasons for this distinct clustering of these four samples compared to the others, we did not consider these four samples as normal representatives of the goat gut microbiome, either because of these goats being sick and having a disruptive gut microbiome or because of contamination.

When removing these four distinctive samples, no significant difference in overall AMR load was identified between the two groups (ResFinder database); however, the road goats had a higher load of functional resistance genes (ResFinderFG database) and novel resistance genes (ResFinderNG). Further investigation would need to be conducted to examine if increased road pollution is associated with higher AMR in the gut microbiome.

Environmental pollutants have been shown to select for xenobiotic degrading functions within the human gut microbiome (4). This is similar to our findings in the Tanzanian goat microbiomes: we found an increased abundance of gene products involved in naphthalene and toluene metabolism in the road goats. Naphthalene is isolated from coal tar fractions, and the main outdoor source is from fugitive emissions and motor vehicle exhaust (77). Toluene is added to gasoline to improve octane ratings but is also used as solvents in, e.g., paints and cleaning agents (78, 92). Similarly, we found one functional ortholog, GST, involved in detoxifications and chemical carcinogenesis being more abundant in road goats. Chemical carcinogens can directly, or post-xenobiotic metabolism, cause cancer by inducing DNA damage. It is widely recognized that exhaust from diesel and gasoline engines contains carcinogenic gasses, particles, volatile and semi-volatile organic compounds (93), and naphthalene is classified as possibly carcinogenic to humans by the International Agency for Research on Cancer (94), which might have similar effects in goats. However, since high abundance of GST did also explain other pathways being enriched in road goats, it can be speculated that it, in this case, is helping the gut bacteria to cope with different toxins from the road exposure. GST is a group of phase II detoxification enzymes with broad specificity, known to protect cells from oxidative stress and xenobiotics by, e.g., conjugating xenobiotics to glutathione to make them more water-soluble and thereby easier to excrete (95).

It was not possible to link the gene orthologs involved in naphthalene or toluene degradation from the road goats to the MAGs, since none of the contigs containing these genes were part of a MAG. However, taxonomic annotation of the contig sequence containing naphthalene or toluene genes showed different *Escherichia* species as being a common host. The *Escherichia* genus was also a common host for glutathione S-transferase (K00799), suggesting that living near heavily trafficked roads will select for xenobiotic

degrading functions within the gut microbiome of goats, and that these functions can be found in, e.g., *Escherichia* species.

The enrichment of GST and cytochrome P450 (CYP450) enzymes in road goats suggests microbiome adaptations to detoxify environmental pollutants. Roadside environments often expose animals to stressors like pollutants, human activity, and dietary shifts due to limited forage diversity and availability. Behavioral responses, such as increased foraging on anthropogenic waste (e.g., plastics), could be a survival strategy. Plastic digestion in urban-grazing ruminants is well documented (96), and plastic waste has been found in the rumen of goats in both DRC, Ethiopia, Somalia, and Nigeria (14–18). Thus, we investigated the goat microbiome ability to degrade plastics. The ability of plastic biodegradation was not enriched in the gut microbiome of road goats, which suggests that living near a busy road does not select for this function within the goat gut microbiome. However, no quantification of environmental pollution from plastic particles was performed in order to characterize to which degree plastic pollution was impacting each location (rural or road).

In conclusion, we have characterized and explored the largely unknown composition of the Tanzanian goat gut microbiomes. We have shown that environmental pollution from a heavily trafficked road in Tanzania alters the gut composition of goats grazing near it, selecting for xenobiotic degrading functions to possibly detoxify these compounds to the host. While there are a number of soil-indicating bacteria in our goat microbiomes, we still could not exclude that some of the observed differences in the functional microbiome between the rural and road samples are originating from contamination from soil bacteria. Future studies should explore the surrounding soil microbiomes to establish the bacterial flow from the soil to the goat microbiomes or vice versa. Our finding aligns with the idea that also goat microbiomes can adapt functionally to environmental conditions, potentially impacting the host's overall health and resilience. Further analysis could explore if this functional enrichment correlates with specific metabolic or health outcomes in road goats.

## ACKNOWLEDGMENTS

We are thankful to Thomas Nordahl Petersen and Christian Brinch for technical assistance. This study was funded by the Global Health EDCTP3 Joint Undertaking (Global Health EDCTP3) program under grant agreement no. 101103059 (GREATLIFE). The funder played no role in study design, data collection, analysis and interpretation of data, or the writing of this manuscript.

E.E.B.J., S.O., T.S., and F.M.A. contributed to the conception and design of the study. T.S. and F.M.A. performed sampling. C.A.S. performed the DNA extraction and sequencing. E.E.B.J. ran the bioinformatics analyses, and M.L.J. contributed. E.E.B.J. carried out the data analyses, created the visualizations, and wrote the manuscript. M.L.J., S.O., T.S., and F.M.A. commented on the manuscript. All authors read and approved the submitted manuscript.

## AUTHOR AFFILIATIONS

[1]Research group for Genomic Epidemiology, National Food Institute, Technical University of Denmark, Lyngby, Denmark
[2]Kilimanjaro Clinical Research Institute, Moshi, Tanzania

## AUTHOR ORCIDs

Emilie Egholm Bruun Jensen  http://orcid.org/0000-0002-3214-7918
Saria Otani  https://orcid.org/0000-0002-2538-8086
Frank M. Aarestrup  http://orcid.org/0000-0002-7116-2723

## FUNDING

| Funder | Grant(s) | Author(s) |
|---|---|---|
| European and Developing Countries Clinical Trials Partnership | 101103059 | Emilie Egholm Bruun Jensen |
| | | Marie Louise Jespersen |
| | | Christina Aaby Svendsen |
| | | Tolbert Sonda |
| | | Saria Otani |
| | | Frank M. Aarestrup |

## AUTHOR CONTRIBUTIONS

Emilie Egholm Bruun Jensen, Data curation, Formal analysis, Investigation, Software, Visualization, Writing – original draft, Writing – review and editing | Marie Louise Jespersen, Software, Writing – review and editing | Christina Aaby Svendsen, Investigation | Tolbert Sonda, Conceptualization, Funding acquisition, Project administration, Writing – review and editing | Saria Otani, Methodology, Supervision, Validation, Writing – review and editing | Frank M. Aarestrup, Conceptualization, Funding acquisition, Methodology, Project administration, Resources, Supervision, Validation, Writing – review and editing

## DATA AVAILABILITY

The datasets generated and analyzed during the current study are available in the European Nucleotide Archive (ENA) repository, https://www.ebi.ac.uk/ena/browser/view/PRJEB83610. This includes PE reads, sequence assembly, as well as the dereplicated MAGs.

The underlying code for this study is available in GitHub and can be accessed via https://github.com/emilieegholmbruun/tanzanian_goat_microbiomes.

## ETHICS APPROVAL

Ethical clearance certificate for conducting research in Tanzania was obtained from the National Institute for Medical Research (NIMR) with reference number NIMR/HQ/R.8a/Vol.IX/4596, and was approved by the Medical Research Coordinating Committee and the Ministry of Health.

## ADDITIONAL FILES

The following material is available online.

### Supplemental Material

**Supplemental material 1 (Spectrum02036-25-S0001.pdf).** Tables S1 to S5; Fig. S1 to S6.
**Supplemental material 2 (Spectrum02036-25-S0002.pdf).** Tables S6 to S9; Fig. S7 to S12.

### Open Peer Review

**PEER REVIEW HISTORY (review-history.pdf).** An accounting of the reviewer comments and feedback.

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
