## [Reviewer comments · Microbiology Spectrum]

Microbiology Spectrum

Tanzanian goat gut microbiomes adapt to roadside pollutants and environmental stressors

Emilie Jensen, Marie Jespersen, Christina Svendsen, Tolbert Sonda, Saria Otani, and Frank Aarestrup

Corresponding Author(s): Emilie Jensen, Danmarks Tekniske Universitet

Review Timeline:

Submission Date:	July 7, 2025
Editorial Decision:	September 8, 2025
Revision Received:	October 20, 2025
Accepted:	November 16, 2025

Editor: Henning Seedorf

Reviewer(s): Disclosure of reviewer identity is with reference to reviewer comments included in decision letter(s). The following individuals involved in review of your submission have agreed to reveal their identity: Kirsten Zwally (Reviewer #2)

Transaction Report:

DOI: <https://doi.org/10.1128/spectrum.02036-25>

Re: Spectrum02036-25 (Tanzanian goat gut microbiomes adapt to roadside pollutants and environmental stressors)

Dear Ms. Emilie Egholm Bruun Jensen:

Thank you for the privilege of reviewing your work. Below you will find my comments, instructions from the Spectrum editorial office, and the reviewer comments.

Revision Guidelines

Sincerely,
Henning Seedorf
Editor
Microbiology Spectrum

Reviewer #1 (Comments for the Author):

The statistical analysis is generally well executed. However, the authors should ensure that the assumptions underlying the tests were verified. In particular, for the independent t-test, the assumption of equal variances is important. From the results presented, it appears that the two groups may have unequal variances for several variables, which could affect the validity of the findings.

Reviewer #2 (Comments for the Author):

This study investigated the differences between goat microbiomes collected from rural versus urban environments. This research is particularly interesting as industrialization and anthropogenic activity can heavily influence an organism's microbiome. This article is very informative and novel. It was a straight-forward read that is digestible for a broad audience. More detail regarding the specifics of sampling would be helpful to understand the total number of goats and samples collected, or a diagram illustrating the methods would be helpful. Although your bioinformatics pipeline is sufficient, consider a flow-chart for ease of reading and visualization. Overall, this article is very interesting and unique. Generating MAGs, although computationally intensive, was a good choice and provided a lot of much needed metagenomic information. The discussion was well thought out with a few minor misspellings and grammatical errors.

More specific comments:

Line 33-34: Reverse the order of abbreviations to metagenome-assembled genomes (MAGs). The full word should be written before the abbreviation.

Line 47: Misspelling "of".

Line 61: et al. does not need to be italicized.

Line 76: Rephrase for clarity (i.e. remove e.g.)

Line 91: Misspelling of "was".

Line 93: Comma needed after "Additionally".

Line 101: Bacterial misspellings.

Sample collection: How were goats selected for sampling (e.g. simple random sampling) or did you use a non-probability sampling method? How many goats were sampled overall from each of the two locations? How were the samples transported back to the lab - wet ice/ dry ice/ liquid nitrogen? Were there any preparations done prior to freezing like glycerol preservatives? Are there any animal welfare/ethics committee regulations for collecting animal samples? For the 34 gut microbiome samples, what number came from road vs rural goats?

Line 150-151: Phrasing is a bit awkward.

Line 172: Too many spaces between words.

Line 174: Assuming the host was goat, did you also remove any human genome contamination?

Line 178: Was beta-diversity included?

Line 196: Too many spaces between "data retrieved".

Statistical Analyses: Would be interested in beta diversity between samples and Bray-Curtis metrics to summarize overall diversity and dissimilarities between rural and road goat microbiomes, but clr and t-test are fitting for the hypothesis.

Bioinformatics: Was assembly quality and completeness compared for MAGs? Genome Standards Consortium (GSC) highly encourages the inclusion of assembly quality/completeness score/contamination scores and software.

Figure 5: Suggestion: it may be helpful to color-code the flows of the Sankey diagram to better correspond to assigned taxa instead of just coloring the nodes. A lot of the colors of the nodes are very similar and difficult to differentiate.

Line 450: Saharan is capitalized.

Overall: Escherichia is misspelt throughout the article.

This paper is exploring the impact of environmental exposure on the gut microbiome of goat in Tanzania. The goat microbiome is not well studied especially in Africa. This is an area more work is need in the microbiome space. The study is good, and the analyses are well done. However, there is need for review to improve the rigor and interpretation of the result.

Major things

1. Consider expanding the introduction and discussion to include current literature on how environmental pollutants affect the microbiome in humans and other species and relate your findings to this broader body of literature.
2. You applied hierarchical clustering to analyze the samples. Could you clarify why this method was chosen instead of more conventional approaches for assessing community structure, such as Bray–Curtis dissimilarity or other distance-based metrics, where differences between rural and roadside goats could be statistically tested?
3. You mapped the reads to chloroplast, and it showed the sample source is goat, did you remove host read from the samples?
4. In your statistical analysis, you compared rural and roadside groups in alpha diversity metrics, total AMR load, and specific gene counts using independent t-tests with Bonferroni correction. Please note that a Bonferroni correction is not necessary when comparing only two groups, as the t-test already controls for Type I error in this case. Additionally, did you verify whether the assumption of equal variances between the two groups was met for all variables analyzed with the independent t-test? From the data presented, it appears that the variances may not be equal, which could affect the reliability of your results.
5. Include more details in your statistical analysis.
6. Did you use any control like mock in your analysis?
7. Include the code you used for your analysis in a public repository like GitHub.

Other minor corrections

1. Line 47: change “og” to “of”.
2. Line 76: I think you should remove the e.g. before human.
3. Line 126: What does “The samples were kept as cool as possible” mean? Can state what you did exactly or how you kept the samples cool?
4. Line130: Should it be “Prior to DNA extraction”?
5. Line 132: DNA was purified or extracted?
6. Line 264: change “MAGs are” to “MAGs were”
7. Line 166 – 173: Let it be clear hear that this is for antimicrobial resistance genes.
8. In Supplementary Figure 4, make it clear that in pane A-C, the distinct samples have been removed. Also, make it clear in other figures legends if the distinct samples were removed.

9. Check the axis title for Supplementary Figure 10 C. You have Glutathione S-transferase in the figure but Toluene in the legend.
10. In Line 397-411, you mentioned Glutathione S-transferase as a cancer pathway, while Glutathione S-transferase is involved, its major function is detoxification, its impairment of ability to detoxify toxic chemicals makes it associated with cancer. For the context of your result, I think you should focus on detoxification.

Overall Recommendation: Modifications

- (1) Are all the authors' conclusions supported by their data? Yes
- (2) Is the manuscript written in standard English and easy to comprehend? Yes
- (3) Does the study include any large datasets that need to be deposited in a public repository?
If the answer is yes, please use the comment box to give us more details
Yes, genomic datasets were deposited into a public repository.
- (4) Does the work described in this study raise any concerns about biosafety or biosecurity that should be discussed prior to publication by the ASM Responsible Publication Committee?
No.
- (5) Have appropriate statistical tests been applied?
Yes

Comments and Suggestions for the Author:

This study investigated the differences between goat microbiomes collected from rural versus urban environments. This research is particularly interesting as industrialization and anthropogenic activity can heavily influence an organism's microbiome. This article is very informative and novel. It was a straight-forward read that is digestible for a broad audience. More detail regarding the specifics of sampling would be helpful to understand the total number of goats and samples collected, or a diagram illustrating the methods would be helpful. Although your bioinformatics pipeline is sufficient, consider a flow-chart for ease of reading and visualization. Overall, this article is very interesting and unique. Generating MAGs, although computationally intensive, was a good choice and provided a lot of much needed metagenomic information. The discussion was well thought out with a few minor misspellings and grammatical errors.

More specific comments:

Line 33-34: Reverse the order of abbreviations to metagenome-assembled genomes (MAGs). The full word should be written before the abbreviation.

Line 47: Misspelling "of".

Line 61: et al. does not need to be italicized.

Line 76: Rephrase for clarity (i.e. remove e.g.)

Line 91: Misspelling of "was".

Line 93: Comma needed after "Additionally".

Line 101: Bacterial misspellings.

Sample collection: How were goats selected for sampling (e.g. simple random sampling) or did you use a non-probability sampling method? How many goats were sampled overall from each of the two locations? How were the samples transported back to the lab – wet ice/ dry ice/ liquid nitrogen? Were there any preparations done prior to freezing like glycerol preservatives? Are there any animal welfare/ethics committee regulations for collecting animal samples? For the 34 gut microbiome samples, what number came from road vs rural goats?

Line 150-151: Phrasing is a bit awkward.

Line 172: Too many spaces between words.

Line 174: Assuming the host was goat, did you also remove any human genome contamination?

Line 178: Was beta-diversity included?

Line 196: Too many spaces between “data retrieved”.

Statistical Analyses: Would be interested in beta diversity between samples and Bray-Curtis metrics to summarize overall diversity and dissimilarities between rural and road goat microbiomes, but clr and t-test are fitting for the hypothesis.

Bioinformatics: Was assembly quality and completeness compared for MAGs? Genome Standards Consortium (GSC) highly encourages the inclusion of assembly quality/completeness score/contamination scores and software.

Figure 2: What software was used for the relative abundance chart created? Was this also Plotly or phyloseq? Spacing is off in figure captions.

Line 334: How was effect calculated?

Figure 5: Suggestion: it may be helpful to color-code the flows of the Sankey diagram to better correspond to assigned taxa instead of just coloring the nodes. A lot of the colors of the nodes are very similar and difficult to differentiate.

Line 450: Saharan is capitalized.

Overall: *Escherichia* is misspelt throughout the article.

Confidential remarks for the Editors:

None at this time.

Reviewer #1

This paper is exploring the impact of environmental exposure on the gut microbiome of goat in Tanzania. The goat microbiome is not well studied especially in Africa. This is an area more work is need in the microbiome space. The study is good, and the analyses are well done. However, there is need for review to improve the rigor and interpretation of the result.

We thank the reviewer for this careful review. We have carefully gone through the comments and adjusted the manuscript accordingly.

Major things

1. Consider expanding the introduction and discussion to include current literature on how environmental pollutants affect the microbiome in humans and other species and relate your findings to this broader body of literature.

We appreciate these suggestions and agree that the manuscript will benefit from this. We have expanded the introduction (lines 107-124) to review the literature on the effect of environmental pollutants on the gut microbiome.

2. You applied hierarchical clustering to analyze the samples. Could you clarify why this method was chosen instead of more conventional approaches for assessing community structure, such as Bray–Curtis dissimilarity or other distance-based metrics, where differences between rural and roadside goats could be statistically tested?

We included both methods as described in the manuscript (Lines 316-320, Figure S3A, Figure 2B). Applying centred log-ratio (CLR) transformation to the microbiome data and then performing ordination analysis (PCA) is similar to Bray-Curtis, only the centred log-ratio takes into account the compositional nature of the dataset. This method is suggested by Gloor et al. (<https://doi.org/10.3389/fmicb.2017.02224>, <https://doi.org/10.1016/j.annepidem.2016.03.003>).

We applied the unsupervised clustering after realizing that some of the samples were completely different in their bacterial composition compared to the rest (e.g., the 4 microbiomes). This method allowed us to objectively identify these as outliers and to verify that they represented distinct community structures rather than normal variation within goat gut microbiomes. We have speculated on the reasons why in the discussion (Lines 512-546).

3. You mapped the reads to chloroplast, and it showed the sample source is goat, did you remove host read from the samples?

We mapped the reads to a mitochondria database (not a chloroplast one) to identify the host. Host reads were not removed since there is no consensus on host removal for metagenomic studies, and only 58% of studies eliminate host contamination (<https://doi.org/10.1093/gigascience/giaf004>). For personal data protection reasons, it might be advisable to remove host DNA in human studies, however this is not applicable in our case. Assembly and binning might benefit from such host removal when considering time and resources (removing host reads can be computationally and source demanding), however this was not a problem in our study. Finally, host reads would not be included in the final MAG's output eventually, therefore we did not opt for this.

4. In your statistical analysis, you compared rural and roadside groups in alpha diversity metrics, total AMR load, and specific gene counts using independent t-tests with

Bonferroni correction. Please note that a Bonferroni correction is not necessary when comparing only two groups, as the t-test already controls for Type I error in this case. Additionally, did you verify whether the assumption of equal variances between the two groups was met for all variables analyzed with the independent t-test? From the data presented, it appears that the variances may not be equal, which could affect the reliability of your results.

We realise that Bonferroni correction is not necessary when only comparing two groups. We have implemented a Levene's test for equality of variances to test this assumption for all comparisons (lines 371-373, lines 408-412, Figure S4, Figure S5, Figure S6, Figure S7, Figure S8, Figure S9, Figure S10, and Figure S12). When applicable we opt for a Welch t-test instead of the standard t-test.

5. Include more details in your statistical analysis.

We have included more details in the statistical analyses (lines 259-262).

6. Did you use any control like mock in your analysis?

Including mock communities is often required when the technology is at the beginning of its adaptation in the laboratory where the experiment is running to control for the machine and procedure detection and accuracy. As running additional mock communities each time will increase the cost largely. This is not the case in our laboratories, where Illumina NovaSeq is used on a regular basis and was accounted for during our QC test in the beginning when the technology was adopted. We have included negative controls (only sterile water) during DNA extractions as negative controls, and those were below DNA detection levels therefore were not able to be sequenced.

7. Include the code you used for your analysis in a public repository like GitHub.

We have added the code used in a GitHub repository and referred to it in the text (lines 635-637).

Other minor corrections

1. Line 47: change "og" to "of".

We have corrected this (line 47).

2. Line 76: I think you should remove the e.g. before human.

We have removed this (line 76).

3. Line 126: What does "The samples were kept as cool as possible" mean? Can state what you did exactly or how you kept the samples cool?

The samples were collected in remote areas where access to electricity is limited. Therefore the samples were kept on a cooling element in the field when collected (like gel packs or simply ice) until brought to the laboratory where they were kept frozen until DNA extractions. The text has been now expanded to increase clarity (Lines 147).

4. Line 130: Should it be "Prior to DNA extraction"?

We have added "to" to the sentence (Line 151).

5. Line 132: DNA was purified or extracted?

We have rephrased this (line 150 and 153).

6. Line 264: change “MAGs are” to “MAGs were”

We have rephrased this (Line 295).

7. Line 166 – 173: Let it be clear here that this is for antimicrobial resistance genes.

We have clarified this (Line 189).

8. In Supplementary Figure 4, make it clear that in pane A-C, the distinct samples have been removed. Also, make it clear in other figures legends if the distinct samples were removed.

The distinct samples have not been removed from A-C, they are included. They are grouped as their own group in D-F. We have clarified this in the figure legend (Figure S4).

9. Check the axis title for Supplementary Figure 10 C. You have Glutathione S-transferase in the figure but Toluene in the legend.

We have corrected this, thanks (legend of Supplementary figure 10C).

10. In Line 397-411, you mentioned Glutathione S-transferase as a cancer pathway, while Glutathione S-transferase is involved, its major function is detoxification, its impairment of ability to detoxify toxic chemicals makes it associated with cancer. For the context of your result, I think you should focus on detoxification

We thank the reviewer for this concern, and we agree the main focus should be on its major function of detoxification. However, we will keep this line of text mentioned as it states the result we observed. In the discussion, we do however speculate that Glutathione S-transferase in our case is helping the gut bacteria to detoxify the toxins from the road exposure and not as part of a cancer pathway. We rephrased this part to put more emphasis on the detoxification part (Lines 572-576).

Reviewer #2

Comments and Suggestions for the Author:

This study investigated the differences between goat microbiomes collected from rural versus urban environments. This research is particularly interesting as industrialization and anthropogenic activity can heavily influence an organism's microbiome. This article is very informative and novel. It was a straight-forward read that is digestible for a broad audience. More detail regarding the specifics of sampling would be helpful to understand the total number of goats and samples collected, or a diagram illustrating the methods would be helpful. Although your bioinformatics pipeline is sufficient, consider a flow-chart for ease of reading and visualization. Overall, this article is very interesting and unique. Generating MAGs, although computationally intensive, was a good choice and provided a lot of much needed metagenomic information. The discussion was well thought out with a few minor misspellings and grammatical errors.

We thank the reviewer for taking the time to read and comment on the manuscript. We have now addressed the comments carefully and adjusted the manuscript accordingly.

More specific comments:

Line 33-34: Reverse the order of abbreviations to metagenome-assembled genomes (MAGs). The full word should be written before the abbreviation.

We have reversed the order (Lines 33-34).

Line 47: Misspelling “of”.

We have corrected this (Line 47).

Line 61: et al. does not need to be italicized.

We realize that et al. is not required to be italicised on many style guides like APA, MLA, and Harvard. However, we will leave it like this.

Line 76: Rephrase for clarity (i.e. remove e.g.)

We have removed this (Line 76).

Line 91: Misspelling of “was”.

We have corrected this (Line 92).

Line 93: Comma needed after “Additionally”.

We have added this (Line 94).

Line 101: Bacterial misspellings.

We have corrected this (Line 102).

Sample collection:

How were goats selected for sampling (e.g. simple random sampling) or did you use a non-probability sampling method?

The sampling was random and based on faecal matter availability in the field. Taken into consideration the attempt to keep the selection balanced between the rural and road area goats. The goats in those remote areas graze freely, therefore the collection happens immediately after defecating in order to keep the sample fresh when possible, this complicates planning a sampling method.

How many goats were sampled overall from each of the two locations?

We have added the numbers of samples collected for each location in the text (Lines 146-147).

How were the samples transported back to the lab – wet ice/ dry ice/ liquid nitrogen?

The samples were collected in remote areas where access to electricity is limited. Therefore the samples were kept on a cooling element in the field when collected when possible (like gel packs or simply ice) until returned to the laboratory where they were kept frozen until DNA extractions. The text has been now expanded to increase clarity (Line 147).

Were there any preparations done prior to freezing like glycerol preservatives?

No preparations were done prior to freezing. The samples were kept in a cool dry place until frozen in the laboratory.

Are there any animal welfare/ethics committee regulations for collecting animal samples?

We thank the reviewer for this point. We collected faecal samples after defecation, this is non-invasive procedure and the animals are not touched or harmed, nor live tissues were sampled, and no research on live animals was conducted. Therefore, no committee or regulations are required for the sampling itself. However, ethical clearance certificate for conducting research in Tanzania are required and acquired from The National Institute for Medical Research (NIMR) with reference number NIMR/HQ/R.8a/Vol.IX/4596, which was approved by The Medical Research Coordinating Committee and The Ministry of Health (lines 598-602). The clearance is described in the Ethical clearance and permission part of the manuscript (Lines 639-643).

For the 34 gut microbiome samples, what number came from road vs rural goats?

We have added this to the “Samples” section in the Methods sections (Lines 146-147).

Line 150-151: Phrasing is a bit awkward.

We have rephrased the sentence (Lines 171-173).

Line 172: Too many spaces between words.

This is a draft line formatting and will not be reflected in the final printing.

Line 174: Assuming the host was goat, did you also remove any human genome contamination?

No, we did not remove human genome contamination, since potential human sequencing reads would not be assembled and binned anyway. It might decrease the assembly- and binning time and resources spent, but we did not have problems with this.

Line 178: Was beta-diversity included?

Yes, beta-diversity was included and described in the method section called “Compositional data analysis”. Here, we use centered log-ratio (clr) transformation to convert the compositional data into a euclidean space, which allows for normal euclidean metrics like distances which we use for clustering in our PCA’s. This follows the method of Gloor et al, since sequencing data is compositional and needs to be handled as such

(<https://doi.org/10.3389/fmicb.2017.02224>,

<https://doi.org/10.1016/j.annepidem.2016.03.003>).

We have changed the section title to reflect that we only include the method for the alpha diversity here (Line 204). We have changed the wording to include “beta-diversity” in the “Compositional data analysis” section of the method (Line 238). Thanks for the suggestion.

Line 196: Too many spaces between “data retrieved”.

This is a draft line formatting and will not be reflected in the final printing.

Statistical Analyses: Would be interested in beta diversity between samples and Bray-Curtis metrics to summarize overall diversity and dissimilarities between rural and road goat microbiomes, but clr and t-test are fitting for the hypothesis.

We are using a compositional data analysis approach as suggested by Gloor et al.

(<https://doi.org/10.3389/fmicb.2017.02224>,

<https://doi.org/10.1016/j.annepidem.2016.03.003>). Sequencing data is compositional and needs to be treated as such. Instead of using the Bray-Curtis dissimilarity measure for beta-

diversity (which is not subcompositionally coherent), they suggest doing centred log ratio transformation (clr) on the raw counts and perform the beta-diversity analysis with compositional biplots (pca). Such analyses are also done in several current microbiome studies (<https://doi.org/10.1080/19490976.2025.2484385>, <https://doi.org/10.1038/s41467-024-51957-8>, <https://doi.org/10.1128/msystems.01018-23>).

Bioinformatics:

Was assembly quality and completeness compared for MAGs? Genome Standards Consortium (GSC) highly encourages the inclusion of assembly quality/completeness score/contamination scores and software.

We agree with the GSC that carefully assessing the quality of the metagenomic assembled genomes is essential. We used CheckM to assess the completeness and contamination of the bins (Line 171-177). We then moved forwards with the high quality (HQ) and medium quality (MQ) bins using the Minimum Information about a Metagenome-Assembled Genome (MIMAG) criteria for defining HQ and MQ bins.

Figure 2: What software was used for the relative abundance chart created? Was this also Plotly or phyloseq? Spacing is off in figure captions.

Only the Sankey diagram was created with the Plotly library. The rest of the figures were created with the python libraries matplotlib.pyplot and seaborn. We have added this to the manuscript (Lines 271-273).

Line 334: How was effect calculated?

Effect is calculated as part of ALDEx2 and is the median difference in clr values between groups divided by the median of the largest difference in clr values within groups i.e. $\text{diff.btw} / \max(\text{diff.win})$ (Lines 256-258).

Figure 5: Suggestion: it may be helpful to color-code the flows of the Sankey diagram to better correspond to assigned taxa instead of just coloring the nodes. A lot of the colors of the nodes are very similar and difficult to differentiate.

We thank the reviewer for this suggestion. We agree that a lot of taxa is represented, but we think it will add more confusion than not coloring the flows as well.

Line 450: Saharan is capitalized.

Saharan is capitalised now (Line 492).

Overall: Escherichia is misspelt throughout the article.

This was corrected throughout the manuscript (Lines 102, 470, 471, 474, 581, 582, 585)

Re: Spectrum02036-25R1 (Tanzanian goat gut microbiomes adapt to roadside pollutants and environmental stressors)

Dear Ms. Emilie Egholm Bruun Jensen:

Your manuscript has been accepted, and I am forwarding it to the ASM production staff for publication. Your paper will first be checked to make sure all elements meet the technical requirements. ASM staff will contact you if anything needs to be revised before copyediting and production can begin. Otherwise, you will be notified when your proofs are ready to be viewed.

The authors provided the link for a GitHub repository for the code used for the analysis but the link is not working. For open and fair science, it is important the authors update the link and make the code available.

Sincerely,
Henning Seedorf
Editor
Microbiology Spectrum

Reviewer #1 (Comments for the Author):

The manuscript is answering a sound question, the study is well deigned, appropriate statistical methods have been applied and results and conclusions are appropriate.

Reviewer #2 (Comments for the Author):

Thank you for the corrections and expansions on this manuscript! Overall, this is a well-rounded study with appropriate analyses that are well-performed and clearly explained. I have no further remarks or suggestions on this manuscript.